# Critical Dynamics of Defects and Mechanisms of Damage-Failure Transitions in Fatigue

**DOI:** 10.3390/ma14102554

**Published:** 2021-05-14

**Authors:** Oleg Naimark, Vladimir Oborin, Mikhail Bannikov, Dmitry Ledon

**Affiliations:** Institute of Continuous Media Mechanics of the Ural Branch, RAS, 614013 Perm, Russia; oborin@icmm.ru (V.O.); mbannikov@icmm.ru (M.B.); ledon@icmm.ru (D.L.)

**Keywords:** fracture, gigacycle fatigue, surface morphology, kinetic equation

## Abstract

An experimental methodology was developed for estimating a very high cycle fatigue (VHCF) life of the aluminum alloy AMG-6 subjected to preliminary deformation. The analysis of fatigue damage staging is based on the measurement of elastic modulus decrement according to “in situ” data of nonlinear dynamics of free-end specimen vibrations at the VHCF test. The correlation of fatigue damage staging and fracture surface morphology was studied to establish the scaling properties and kinetic equations for damage localization, “fish-eye” nucleation, and transition to the Paris crack kinetics. These equations, based on empirical parameters related to the structure of the material, allows us to estimate the number of cycles for the nucleation and advance of fatigue crack.

## 1. Introduction

The evaluation of the longevity of critical engineering structures, for example, working parts of aircraft engines, raises new fundamental problems associated with the calculation of service reliability and operating time of materials under cyclic loading during more than 10^9^–10^10^ cycles, generally known as a gigacycle fatigue range [1].

Two problems of a fundamental nature determine the characteristics of fatigue failure at VHCF [1,2]: the problem of localization of damage in conditions of multiscale nucleation and growth of defects, and the problem of the propagation of cracks in the damaged material.

The difficulties in separating the stages of the fatigue crack origin and propagation initiated a fundamentally new problem: a general formulation of damage failure staging including the multiscale kinetics of damage localization, nucleation, and growth of the fatigue crack [3,4,5]. The duration of these stages is essentially determined by the state of the material structure and the kinetics of the free energy release, revealing the features of critical phenomena [3]. In contrast to HCF, when the fatigue life is associated with the crack advance, the scenario of VHCF damage-failure transition arises in the course of multiscale defects kinetics (PSB, microcracks, pores). Moreover, VHCF reveals a qualitative difference in the nucleation of fatigue cracks in the material, leading to reformulation of the fatigue life problem in which the critical damage-failure transition is estimated with reference to the nonlinearity of the free energy release. The role of the initiation stage is especially important for VHCF loading, which is characterized by the emergence of a “fish-eye” area, the development of small cracks with structure sensitive kinetics that is qualitatively different from the kinetics of crack growth for HCF associated with the Paris law. The correspondence of damage kinetics, the free energy release nonlinearity and specific morphological pattern of the fracture surface [6,7] are the key questions to identifying the VHCF damage-failure transition staging [8,9] discussed in the paper.

Within this fatigue load range, of considerable practical interest is the range corresponding to the number of cycles *N* ≈ 10^9^ [3]. The behavior of materials in this range is determined by qualitative changes, mechanisms responsible for the initiation and growth of the cracks.

The process of material fracture in the range of gigacycle loading involves several stages, which are classified on account of the damage-induced structural changes occurring on different spatial scales including persistent slip bands (PSBs), fatigue striations, microcracks (formed as a result of PSB crossing), and grain-boundary defects [10,11]. The primary failure is associated with defect scales in the range of 0.1 μm–1 mm, which are considerably smaller than those detected by the standard nondestructive testing methods widely used to assess the fatigue strength of a structure during high cycle fatigue loading.

Today, the quantitative fractography technique is widely used to investigate the role of initial structural heterogeneity, monitoring of defect accumulation at different scales (dislocation ensembles, micropores, microcracks), and determining critical conditions for the damage-failure transition. This method has allowed researchers to distinguish the characteristic stages of fracture (crack nucleation and propagation) and to evaluate the fatigue life of materials and structures under gigacycle loading.

The description of the fracture surface morphology in terms of spatial temporal invariants was first proposed by Mandelbrot [12,13]. This approach is based on the analysis of the fracture surface relief, showing the property of self-affinity, which manifests itself as the invariance of the surface relief characteristics over a broad spectrum of spatial scales. On the other hand, these characteristics reflect the interrelation of defects at different scales evolving into the stages of damage-failure transition and fatigue crack propagation.

The unified laws, governing the relationship between the crack growth rate and a change in the stress intensity factor, are the focus of recent theoretical and experimental studies. The power law, which was first deduced by Paris [1], establishes the subordination of the fatigue crack advance to the self-similar nature of damage accumulation in the vicinity of the crack tip (the process zone). The Paris crack kinetics *da**/dN* (a is the crack length, *N* is the number of cycles) obeys the power law with regard to the magnitude of the stress intensity factor, expressed as $\Delta K=K_{max}-K_{min}$, where *K_max_* and *K_min_* are the maximum and minimum values of stress intensity factors under conditions of cyclic loading:(1)$$\frac{da}{dN}=C{\left(\Delta K\right)}^{m}$$
where *C* and m are dependent material constants. For many materials and different crack growth rates under conditions of high cycle fatigue (HCF), the power exponent m is close to 2–4. The fracture of the material under fatigue loading in the Paris regime is defined by the applied stresses and the length of the original crack (and its orientation). In contrast, in the case of small cracks or small stresses, when the structure and damage of the surrounding material are the main contributing factors to the kinetics of crack growth, the traditional formulation of the Paris law needs considerable revision [1].

## 2. Materials and Methods

The successive dynamic and fatigue tests are of particular importance for predicting fatigue strength of gas turbine materials in the conditions of the so-called foreign object damage (FOD) [14]. In [5], the specimens made of aluminum AlMg6 were subjected to dynamic loads using the split Hopkinson pressure bar (SHPB) machine (Perm, Russia) at the strain rate of ~10^3^ s^−1^ and then subjected to cyclic loading on a Shimadzu USF-2000 ultrasonic loading machine (Tokyo, Japan) at room temperature. Table 1 shows the chemical constitution (percentage by weight) of the AlMg6 alloy. The SHPB machine includes a gas gun and three cylindrical bars (Figure 1), known as the striking bar (SB), incident bar (IB), and transmission bar (TB). The gas gun is used to accelerate the striking bar, which transfers the shock wave into the IB.

The compressive pulse, which passes through the holder and the specimen, does not produce plastic deformation in the specimen (the main impact of the propagating wave is on the holder, which because of high plastic yield strength shows considerable resistance to the damaging action of the incident bar). As soon as the compressive pulse reaches the free end of this bar, it is reflected as a tensile shock wave, which is just the initial incident wave that causes stretching of the specimen. A part of the tensile pulse after reaching the specimen is transferred through it into the first bar, while another part of this pulse moves back into the second bar. Eventually, this results in plastic deformation of the part of the specimen, which is adjacent to the smallest cross section; the holder, which is disconnected with the bars, does not experience tensile stress.

Then, we performed a series of fatigue tests for 14 initial and nine dynamic preloaded specimens using the Shimadzu USF-2000 ultrasonic fatigue testing machine, which provides cyclic loading (R = −1) [2]. The specimens were stressed by a generator, which transformed the frequency of 50 Hz into an ultrasonic electrical sinusoidal signal of frequency 20 kHz with the aid of a piezoelectric transducer. The latter generates longitudinal ultrasonic waves in the frequency range of 20 kHz and the mechanical stress with maximum amplitude at the center of the specimen. The geometry of the specimens is shown in Figure 2. Compressed air cooling of the specimens was used.

The 0.5 kHz difference in the frequency, which was due to approaching the stage of critical damage, could be viewed as a failure precursor associated with the formation of a crack with a characteristic size of ~2 mm. The applied stresses ranging from 110 to 162 MPa made it possible to investigate the fatigue life under gigacycle loading up to 10^10^ cycles. The results of the fatigue testing are shown in Figure 3.

We observed that the cyclic life on the base of 10^9^ cycles for the preloaded AlMg6 alloy decreased from the stress level of 162 MPa to 121–138 MPa (15–25%). There were two specimens (No. 1 and No. 2) that formed fatigue cracks inside the bulk with a characteristic type of fracture called “fish eye”. The other ones fractured from the surface.

To determine the mechanism of internal crack formation, the method of amplitude-frequency analysis of changes in the effective elastic properties of materials was used as proposed in [15,16] on the basis of acoustic properties. The resonance testing machine (Shimadzu USF-2000) was combined with a highly sensitive inductive sensor and an analog-to-digital converter system (Figure 4) that measured the amplitudes and frequencies of oscillations of the free end face of the specimen. The software reads the 65,536-point signal every 0.001 s and performs fast Fourier transform to determine the values of fundamental frequency, second and third harmonics and their amplitudes, and wrote them to a file directly during the experiment. By changing the value of these amplitudes, the behavior of the defective structure of the material can be estimated using the nonlinear parameter of the signal *β* [15,16].

The basis of the damage staging analysis is the registration of current specimen impedance in the presence of harmonic components: the frequency ω_0_, associated with forced vibration amplitude A_1_ and the amplitude A_2_ of the second harmonics with a frequency 2ω_0_, and higher harmonics associated with the influence of defects.

The nonlinearity coefficient *β* [15,16] is directly related to the amplitude of the second harmonic. Crack formation and growth lead to a significant increase in the nonlinearity parameter *β* [16]. In this study, we did not need to know the specific value of the nonlinearity parameter, only its qualitative change in time, so it was enough to measure only the amplitude of the second harmonic. The value of amplitude was measured by an inductive displacement sensor with a signal recording frequency of 10 MHz (Figure 4).

To describe the kinetics of crack growth for sizes smaller than the size of “Paris cracks”, a phenomenological relation was proposed in [17,18], which along with the macroscopic characteristic of the stress state at the crack tip included the structural parameters of the Burgers vector b and the effective stress intensity factor (Figure 5). The authors of [19] proposed structural parameters *l_sc_* (the scale of interaction of defects) and *L_pz_*, (size of the process zone), which are determined from the fracture surface profile using the correlation function [20,21,22].

The self-similar patterns of the nature of fatigue crack growth on specimens loaded in the high and very high-cycle fatigue regimes were studied using the methods of similarity and dimension theory, and as a result, the authors in [21] proposed an equation for crack growth rate, taking into account the structural parameters:(2)$$\frac{da}{dN}=l_{sc}{\left(\frac{\Delta K}{E\sqrt{l_{sc}}}\right)}^{\alpha }{\left(\frac{L_{pz}}{l_{sc}}\right)}^{\beta }$$
where *α* and *β* are the power exponents reflecting the intermediate asymptotic nature of the crack growth kinetics as a function of dimensionless variables $\Delta K_{eff}/E\sqrt{l_{sc}},l_{sc},L_{pz}$. The parameter $\Delta K_{eff}=\Delta K{\left(L_{PZ}/l_{sc}\right)}^{a/b}$ is introduced, which allows us to write Equation (2) in a form similar to the Paris law:(3)$$\frac{da}{dN}=l_{sc}{\left(\frac{\Delta K_{eff}}{E\sqrt{l_{sc}}}\right)}^{\alpha }$$

Equation (3) can be used to describe both small and large cracks, the kinetics of which are determined by the structural parameters *l_sc_, L_pz_*, and scaling indices *α*, *β*. Parameters *l_sc_* and *L_pz_*, associated with defects, can be determined by fractography of the fracture surface (Figure 6).

Integrating (3) and setting $\Delta K=\Delta \sigma \sqrt{\pi a}$, we obtain the expression for the number of cycles required for the growth of a fatigue crack from length *a*_1_ to *a*_2_:(4)$$N=\frac{2\left(a_{1}^{1-\frac{\alpha }{2}}-a_{2}^{1-\frac{\alpha }{2}}\right)}{\alpha -2}\frac{{\left(\frac{L_{pz}}{l_{sc}}\right)}^{-\beta }\pi^{\frac{-\alpha }{2}}{\left(\frac{\Delta \sigma }{E\sqrt{l_{sc}}}\right)}^{-\alpha }}{l_{sc}}$$

## 3. Results

To determine the exact number of cycles required to achieve a certain crack length, we used the methodology to analyze the nonlinear dynamics of oscillations of the free end of the sample described in [15,16,17]. Three characteristic areas can be distinguished on the fracture surface: 1—fine granular area (FGA); 2—the “fisheye” zone; and 3—the crack growth zone (Figure 6). Sizes of the respective regions were measured using the Hirox optical microscope.

For specimen No. 1 (Figure 6a), the following values of crack length (radius) were taken: zone 1—*a*_0_ = 152 μm, zone 2—*a_i_* = 270 μm, and zone 3—*a_k_* = 2679 μm. In all three zones (Figure 6), using the New-View 5010 interferometer-profiler and the procedure described in [13,20], the *l_sc_* and *L_pz_* values were determined on one-dimensional profiles whose directions coincided with the crack propagation. The morphology of fractured surfaces was investigated using an optical interferometer New-View 5010 (Middlefield, CT, USA) (magnification ×2000), which allows digital three-dimensional surface profiles to be obtained. The interaction of defects with each other during crack formation and growth should leave traces on the fracture surface, and if they exhibit correlated behavior in a wide range of scales, this will be reflected in the form of fractal patterns [19,20,21].

The surface was scanned near the crack nucleation site (Figure 6a). One-dimensional surface profiles were investigated in the directions of crack growth, starting from zone 1 to zone 3. The profile resolution during scanning was 0.1 nm in the vertical direction and ~0.5 μm in the horizontal direction.

From these profiles, the scale-invariant Hurst parameter [22] was determined, which is calculated from the slope of the correlation function:(5)$$K\left(r\right)={\langle (z\left(x+r\right)-z\left(x\right))}^{2}\rangle {}_{x}^{1/2}\propto r^{H}$$
where *K(r)* is the average difference between the surface elevation values *z(x + r)* and *z(x)* in a window of size *r*, and *H* is the Hurst index (surface roughness index).

The representation of the function *K(r)* in logarithmic coordinates (Figure 7), which should be linear, allowed us to estimate the lower bound of the scaling-scale *l_sc_*, and to consider the value of the upper bound as the characteristic scale of the process zone *L_pz_*—the area of the correlated behavior of multiscale defect structures.

The values of *l_sc_* and *L_pz_* were determined. For sample No. 1: for zone 1: *l_sc_* = 0.8 μm, *L_pz_* = 11.6 μm; for zone 2: *l_sc_* = 2.2 μm, *L_pz_* = 28.4 μm; and for zone 3: *l_sc_* = 0.4 μm, *L_pz_* = 16.6 μm. For sample No. 2: for zone 1: *l_sc_* = 0.6 μm, *L_pz_* = 17.2 μm; for zone 2: *l_sc_* = 0.9 μm, *L_pz_* = 26.3 μm; and for zone 3: *l_sc_* = 0.5 μm, *L_pz_* = 26.2 μm.

The appearance of a fatigue crack caused a major change in the amplitude of the second harmonic and the crack growth, respectively, its monotonic increase (Figure 8). On the basis of data measuring the amplitude of the second harmonic in real time during the fatigue tests, the number of cycles was determined, which was spent on the nucleation and growth of the fatigue crack. The number of cycles required for the nucleation of the fracture site (the first peak in Figure 8) was *N*_1_ = 7.43∙× 10^8^. The number of cycles during which the crack grew in zone 2 (Figure 6) was the time between the first and second peaks in Figure 8 (i.e., approximately *N*_2_ = 1.4∙× 10^6^ cycles). The remaining *N*_3_ = 6.6∙× 10^6^ cycles were the crack growth time in zone 3.

Then, in Equation (4), two unknown constants remain: *α* and *β*. Their values were determined in the process of solving the minimization problem between the experimental and theoretical number of cycles required for crack growth from 0 to *a*_0_ (zone 1), from *a*_0_ to *a*_i_ (zone 2), and from *a*_i_ to *a_k_* (zone 3), respectively. The values of *α* and *β* were as follows: zone 1: *α* = 2.82, *β* = 0.45; zone 2: *α* = 3.56, *β* = 1.19; zone 3: *α* = 6.22, *β* = 1.35.

The kinetic diagram built according to Equation (3) for a fatigue crack from size *a*_0_ to *a_k_* is shown in Figure 9a for sample No. 1 and in Figure 9b for sample No. 2.

Figure 9 shows that the proposed approach allows us to describe the effect of a change in the growth rate of a fatigue crack upon transition from one characteristic region to another. The constants determined for one of the samples (sample No. 1) gave a satisfactory prediction of the number of cycles to failure for other samples. For example, for sample No. 2, the experimental number of cycles to failure was 7.82 × 10^8^, and the predicted was 6.92 × 10^8^ (relative error of 11%). The calculation results for sample No. 2 with constants α and β obtained on sample No. 1 gave qualitatively consistent results on the number of cycles that went into nucleation and crack growth: 6.91∙× 10^8^ to nucleation; 0.56∙× 10^6^ for growth in zone 2; and 0.58∙× 10^6^ for growth in zone 3. The simulation of the behavior of the second harmonic during the experiment was carried out in [23].

## 4. Discussion

The prediction of the VHCF life time, which includes the definition of fatigue failure as the damage-failure transition problem can be studied by the staging of the multiscale damage localization kinetics, nucleation, and growth of the fatigue crack. The duration of these stages is essentially determined by the state of the material structure that is crucially important for the reliability prediction of the fan blades in the situation of the “foreign object damage”. In contrast to HCF, when the fatigue life is associated with the crack advance, the scenario of VHCF damage-failure transition arises in the course of multiscale defects kinetics (PSB, microcracks, pores). At the same time, VHCF reveals a qualitative difference of fatigue crack initiation in the bulk of the material that leads to reformulation of the fatigue life assessment problem as critical damage-failure transition related to the nonlinearity of the free energy release. The role of the initiation stage is especially important for VHCF loading, revealing the specific surface pattern as the “fisheye” area, and the development of small cracks with structure sensitive kinetics that are qualitatively different from the kinetics of crack advance for HCF associated with the Paris law. The correspondence of damage kinetics, the free energy release nonlinearity, and specific morphological pattern of the fracture surface are the key questions to identify the VHCF damage-failure transition staging.

Statistical theory of defects allowed for the development of the phase field approach of damage-failure transition based on specific nonlinear presentation of the free energy release [24,25]. The damage-failure transition staging is linked to the defect induced relaxation properties associated with the generation of collective modes of defects: solitary slip modes and blow-up modes. These modes have the nature of self-similar solutions reflecting singularity features of characteristic morphology areas on the fatigue fracture surface. A special technique was proposed to identify the damage staging during “in situ” measurement of the acoustic impedance of the sample at VHCH loads. Additionally, in classical singularity of the stress field at the crack tip (the stress intensity factor), the presence of singularities of damage accumulation leads to the spatial-temporal kinetics of characteristic areas on the fatigue fracture surface (fisheye, FGA) with pronounced scaling properties of defect induced roughness. Duality of singularities at the fatigue crack process zone (solitary wave PSB image and the stress singularity) explains the power law universality of the Paris law related to the anomaly of the energy absorption in the PSB areas. The self-similar features of damage kinetics were reflected in kinetic equations with material parameters estimated by scaling analysis of the fracture surface. These equations were used for the prediction of the fatigue life-time at the consecutive loading.

The experimental methodology was developed for estimating the ultra-high cycle lifetime with reference to the situation of accidental high-speed collision of solid particles with fan blades and subsequent fatigue failure in the flight cycle conditions, which is common in the practice of operating aircraft engines. We estimated that the fatigue limit of the dynamically preloaded AlMg6 alloy decreased from the stress level of 162 MPa to 121–138 MPa (15–25%), which corresponds to the critical number of cycles ~10^9^.

## 5. Conclusions

VHCF conditions have both fundamental and applied issues related to qualitative new aspects of fatigue life prediction due to pronounced damage accumulation staging. The definition of stages of initiation and propagation of fatigue cracks is one of the key problems of fatigue failure that can be analyzed by methodology that combined the experimental technique linking the structural and mechanical aspects with the nonlinearity of damage-failure transition. Critical defect kinetics allows one to establish the link of damage staging, fracture surface morphology with self-similar laws of damage kinetics, and fatigue crack advance. Original experiments and scaling analysis of fracture surface were proposed for the formulation of damage kinetic equations and fatigue crack advance.

Summarizing the results, the following conclusions concerning the staging of damage-failure transition in VHCF can be proposed:(I)Characteristic stages of VHCF damage-failure transition follow the qualitative different mechanisms of crack initiation, crack growth, and crack advance related to the nonlinearity of the free energy release;(II)There are pronounced quantitative differences of the fracture surface pattern for the initiation, crack origin, and crack advance related to different scaling properties and nonlinearity of the damage induced free energy release.

## Figures and Tables

**Figure 1 materials-14-02554-f001:**
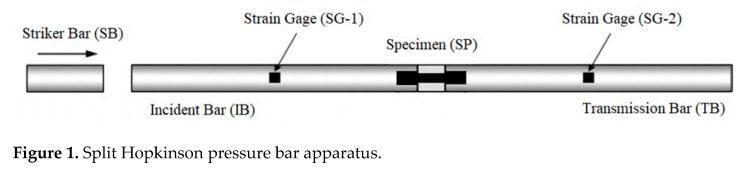
Split Hopkinson pressure bar apparatus.

**Figure 2 materials-14-02554-f002:**
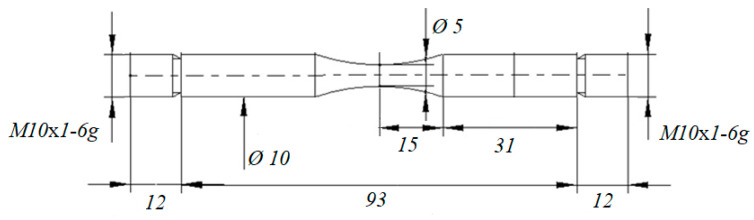
The specimen geometry used for dynamic preloading and cycle tests (sizes in mm).

**Figure 3 materials-14-02554-f003:**
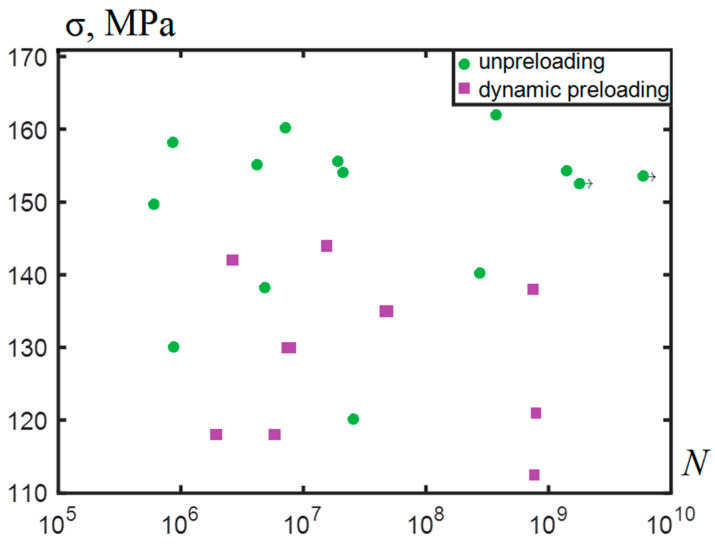
The *S–N* curve (Stress in MPa) in the case of dynamic preloading and in the absence of preloading for the AlMg6 alloy.

**Figure 4 materials-14-02554-f004:**
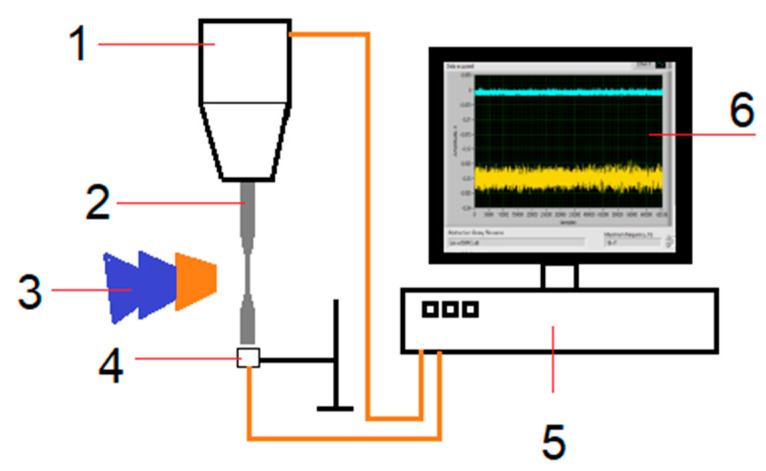
The experimental setup: 1—horn, 2—specimen, 3—cooling system, 4—displacement sensor, 5—controlling and analog-digital converter system, 6—analyzing software.

**Figure 5 materials-14-02554-f005:**
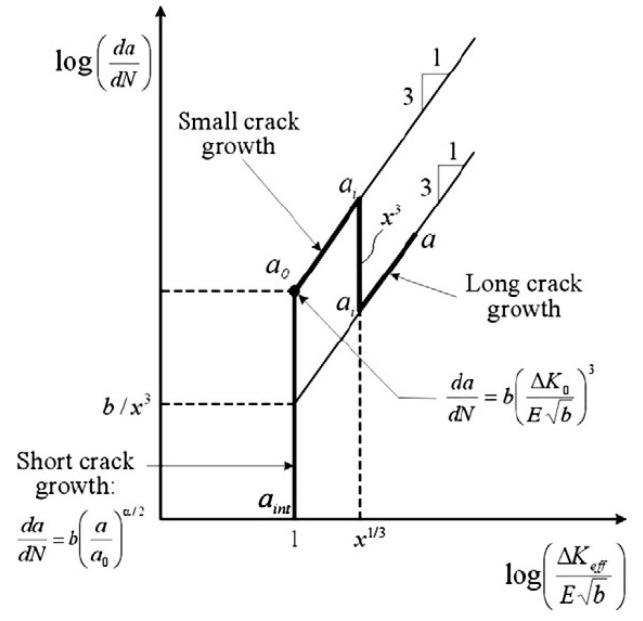
Crack advance diagram in HCF [17,18]: b is the Burgers vector, $\Delta K_{0}$ and $\Delta K_{eff}\;$ are the stress intensity factors corresponding to the crack lengths *a*_0_ and *a_i_*.

**Figure 6 materials-14-02554-f006:**
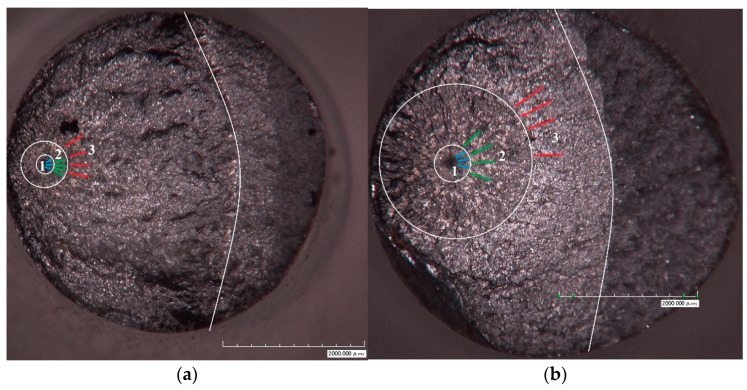
(**a**) The fracture surface of specimen No. 1 (σ = 138 MPa, *N* = 7.51 × 10^8^). (**b**) The fracture surface of specimen No. 2 (σ = 120 MPa, *N* = 7.82 × 10^8^). Colored lines illustrate the directions of analyzed profiles in zones 1 (blue), 2 (green), and 3 (red).

**Figure 7 materials-14-02554-f007:**
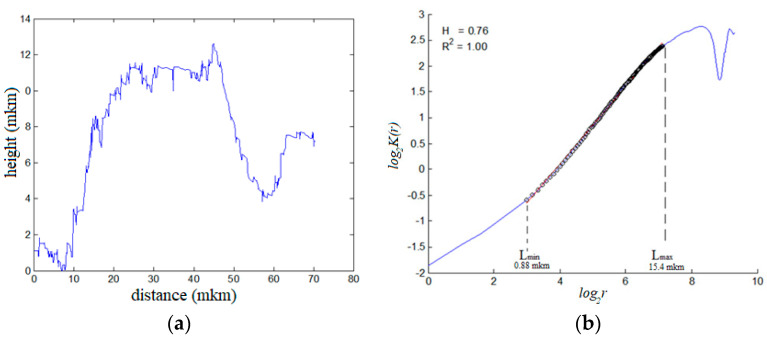
Characteristic for “fisheye” zone 1 of sample No. 2: (**a**) one-dimensional profile, (**b**) plot *log*_2_*K*(*r*) vs. *log*_2_(*r*).

**Figure 8 materials-14-02554-f008:**
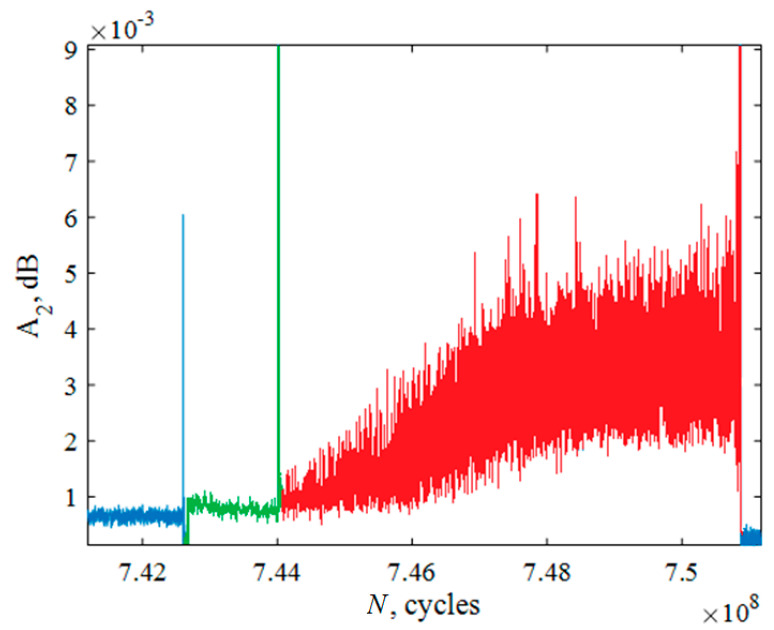
The amplitude of the second harmonic for sample No. 1. Blue color (*N* = 7.426∙× 10^8^ cycles)—damage accumulation and nucleation of fatigue crack site. Green color (*N* = 1.4∙× 10^6^ cycles)—crack grew inside zone 2, formation of “fisheye”. Red color (*N* = 6.7∙× 10^6^)—crack grown by Paris law.

**Figure 9 materials-14-02554-f009:**
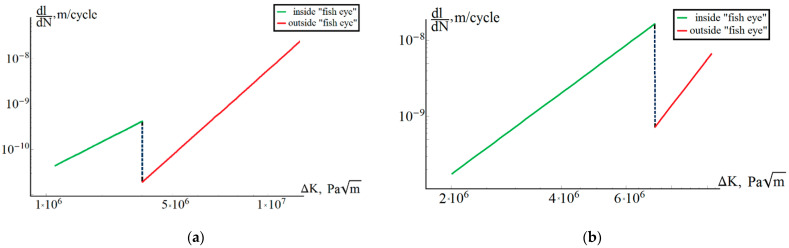
(**a**) Kinetic diagram for sample No. 1. (**b**) Kinetic diagram for sample No. 2.

**Table 1 materials-14-02554-t001:** Chemical composition of AlMg6 (percentage by weight).

Al	Si	Fe	Cu	Mn	Mg	Zn	Ti	Be
92.1	0.24	0.32	0.10	0.7	6.44	0.024	0.035	0.0006

## Data Availability

All data generated or analyzed during this study are included in the article.

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
