# Peer review of "Critical Dynamics of Defects and Mechanisms of Damage-Failure Transitions in Fatigue"

_materials, 2021, doi:10.3390/ma14102554_

Round 1
Reviewer 1 Report
This paper proposes an experimental methodology for estimating a Very High Cycle Fatigue life of aluminum alloy AMG-6 subjected to preliminary deformation. However, there are some problems in this paper that need to be modified. Thus, it needs a major revision according to the following comments:
- The introduction part of the paper lacks a summary of previous research work which are similar to your work.
- The materials and methods part of the paper is not clear enough, and the specific setups of the experience need to be clearly specified.
- References in the paper are cited too carelessly, such as references 1 to 9.
- How many sample used for dynamic preloading and cycle tests? Whether the error object is taken into account.
- The results section has a lot of content that should belong to methods and discussions
- The discussion and conclusions of the paper are not rich enough, so it is suggested to add more content.
- Some details of the paper need to be noticed, such as the writing of professional terms

Author Response
The introduction part of the paper lacks a summary of previous research work which are similar to your work.
- The introductory part of the article has been edited to clarify the purpose of the work and the novelty of the research
The materials and methods part of the paper is not clear enough, and the specific setups of the experience need to be clearly specified.
-
- The part on methods and materials has been revised for clearly specified: we added figure 3 with S-N curve of fatigue test, clarify operating principle of software, added schematic images of investigated profiles on figure 6.
References in the paper are cited too carelessly, such as references 1 to 9.
-
- The introduction part has been revised and the purpose of the references is described in more detail.
How many sample used for dynamic preloading and cycle tests? Whether the error object is taken into account.
The figure 3 with S-N curve of fatigue test was added with number of tested specimens.
The results section has a lot of content that should belong to methods and discussions
The discussion and conclusions of the paper are not rich enough, so it is suggested to add more content.
- The discussion part was supplemented with a conclusion to specify the results of the work:
«Summarizing the results the following conclusion concerning the staging of damage-failure transition in VHCF can be proposed:
(I) characteristic stages of VHCF damage-failure transition follow to qualitative different mechanisms of crack initiation, crack growth and crack advance related to the nonlinearity of the free energy release;
(II) there are pronounced quantitative differences of the fracture surface pattern for the initiation, crack origin and crack advance related to different scaling properties and nonlinearity of the damage induced free energy release.»
Some details of the paper need to be noticed, such as the writing of professional terms
-
- Text of the article has been edited and the terms have been brought into line with their meanings

Reviewer 2 Report
The article is valuable for the science . I propose publication in present form.
Author Response
- The introductory part of the article has been edited to clarify the purpose of the work and the novelty of the research
- Text of the article has been edited and the terms have been brought into line with their meanings
- The part on methods and materials has been revised for clearly specified: we added figure 3 with S-N curve of fatigue test, clarify operating principle of software, added schematic images of investigated profiles on figure 6.
- The discussion part was supplemented with a conclusion to specify the results of the work:
«Summarizing the results the following conclusion concerning the staging of damage-failure transition in VHCF can be proposed:
(I) characteristic stages of VHCF damage-failure transition follow to qualitative different mechanisms of crack initiation, crack growth and crack advance related to the nonlinearity of the free energy release;
(II) there are pronounced quantitative differences of the fracture surface pattern for the initiation, crack origin and crack advance related to different scaling properties and nonlinearity of the damage induced free energy release.»
Reviewer 3 Report
The paper deals with the prediction of the Very High Cycle Fatigue Behaviour of an aluminium alloy subjected to preliminary deformation. The paper is well written and the experimental methodology proposed by the authors is well explained. No major nor minor issues were emerged by revising the manuscript therefore it can be accepted in the present form.
Author Response

(The authors gave the same response as above.)

Reviewer 4 Report
Interesting contribution. Some missing blankets between words. For example the labeling of Fig. 6: "Characteristic for".
Author Response

(The authors gave the same response as above.)

Reviewer 5 Report
The manuscript submitted for review entitled “Critical Dynamics of Defects and Mechanisms of Damage-Failure Transitions in Fatigue” concerns the analysis of fatigue damage staging at VHCF tests.
From the analysis of the information presented in this manuscript, I found the following:
- The introductory part of the manuscript is very general - it should be appropriately enriched.
- The most important information about the content of the manuscript should also be found in the introductory section.
- What is the significant scientific novelty that is the subject of publication? This information is missing in the introduction section.
- line 53: editing error - please correct it.
- lines 66-67: lines 66-67: what does 103 s ^ (- 1) mean? Shouldn't this be corrected to 10^3 s ^ (- 1)?
- the terms "sample" and "specimen" are used interchangeably throughout the body of the manuscript. These terms are not identical in language and cannot be used alternately.
- "The longitudinal ultrasonic waves at a frequency 20 kHz and the mechanical stress with maximum amplitude in the gauge length (mid-crossection) of the sample." - I do not understand what information this phrase was supposed to convey, please correct it or clarify it.
- "Deviation of the frequency" (line 97) - what frequency is it about? Forcing frequency? system response frequency? I am asking for an appropriate explanation of this aspect, as it confuses the reader.
- Line 105-107 - please enter a graphic diagram into the manuscript describing the operation of the developed measurement software.
- Paragraph (lines 185-192) - what is the quantity related to the dimension? Is it distance, radius / diameter? Please clarify this issue.
- Paragraph (lines 193-197)Information on the position of the line profiles in the fisheye zone should be completed with an appropriate drawing.
- The number of cycles required for the nucleation of the fracture site (the first peak in Fig. 7) was N1 = 7.43∙108. - Please explain in detail how the first peak in fig. 7 (blue line) ~ 5.415 x 10 ^ 4 s corresponds to 7.43x 10 ^ 8 cycles? This is a very incomprehensible part of the manuscript and needs improvement.
- The conclusions from the presented manuscript are very general and require a thorough change. The conclusions do not include a qualitative and quantitative summary of the presented experiments.
Based on the above comments, I believe that the manuscript in its current form cannot be published and requires many editorial corrections and substantive supplements.
Author Response
- The introductory part of the manuscript is very general - it should be appropriately enriched.
The most important information about the content of the manuscript should also be found in the introductory section.
What is the significant scientific novelty that is the subject of publication? This information is missing in the introduction section. - The introductory part has been revised
- line 53: editing error - please correct it.
- Corrected
- lines 66-67: lines 66-67: what does 103 s ^ (- 1) mean? Shouldn't this be corrected to 10^3 s ^ (- 1)?
- Corrected
- the terms "sample" and "specimen" are used interchangeably throughout the body of the manuscript. These terms are not identical in language and cannot be used alternately.
- Text of the article has been edited and the terms have been brought into line with their meanings
- "The longitudinal ultrasonic waves at a frequency 20 kHz and the mechanical stress with maximum amplitude in the gauge length (mid-crossection) of the sample." - I do not understand what information this phrase was supposed to convey, please correct it or clarify it.
- This sentence has been edited to clatify. The specimens were stressed by a generator which transformed the frequency of 50 Hz into ultrasonic electrical sinusoidal signal of frequency 20 kHz by piezoelectric transducer producing longitudinal ultrasonic waves at a frequency 20 kHz and the mechanical stress with maximum amplitude in the center of the specimen.
- "Deviation of the frequency" (line 97) - what frequency is it about? Forcing frequency? system response frequency? I am asking for an appropriate explanation of this aspect, as it confuses the reader.
- It is resonant frequency. The difference in the frequency of 0.5 kHz was due to reaching the critical damage stage and considered to be a failure precursor associated with the formation of a crack with characteristic size of ~2 mm.
- Line 105-107 - please enter a graphic diagram into the manuscript describing the operation of the developed measurement software.
- We used the standard package for the fast Fourier transform of the LabView platform, so if we draw a block diagram, it will consist of four rectangles and will not carry interesting information. The sentence was reformulated as follows: «The software read the 65536-point signal every 0.001 seconds and performed fast Fourier transform to determine the fundamental frequency, second and third harmonics and their amplitudes, and wrote it to a file directly during the experiment.»
- Paragraph (lines 185-192) - what is the quantity related to the dimension? Isitdistance, radius / diameter? Pleaseclarifythisissue.
- It is radius, Explanation included in the text of the article.
- Paragraph (lines 193-197)Information on the position of the line profiles in the fisheye zone should be completed with an appropriate drawing.
- The authors made changes in Figure 6, the images of the profiles and their directions within the zones are shown.
- The number of cycles required for the nucleation of the fracture site (the first peak in Fig. 7) was N1 = 7.43∙108. - Please explain in detail how the first peak in fig. 7 (blue line) ~ 5.415 x 10 ^ 4 s corresponds to 7.43x 10 ^ 8 cycles? This is a very incomprehensible part of the manuscript and needs improvement.
- Thank you for your comment. Initially, time was displayed on the x-axis in Figure 8, but the reviewer rightly noted that it does not represent the number of test cycles very well. The fact is that the countdown began earlier than the start of the fatiguetests, so there was a mismatch. The data has been synchronized and for ease of perception, the graph now shows the number of cycles on the x-axis in the article.
- The conclusions from the presented manuscript are very general and require a thorough change. The conclusions do not include a qualitative and quantitative summary of the presented experiments.
- The discussion part was supplemented with a conclusion to specify the results of the work:
«Summarizing the results the following conclusion concerning the staging of damage-failure transition in VHCF can be proposed:
(I) characteristic stages of VHCF damage-failure transition follow to qualitative different mechanisms of crack initiation, crack growth and crack advance related to the nonlinearity of the free energy release;
(II) there are pronounced quantitative differences of the fracture surface pattern for the initiation, crack origin and crack advance related to different scaling properties and nonlinearity of the damage induced free energy release.»

Round 2
Reviewer 1 Report
After last revision, the quality of the paper has been greatly improved, but there are still problems that have not been properly addressed. Thus, it needs a major revision again.
- Question2 is not properly addressed. The specific setups of the experience need to be clearly specified.
- Whenthis passage studies the fatigue performance of the workpiece, does the influence of the material and defect of the workpiece been taken in consideration?
- References citation 1 to 9 are still exist in your paper. (Page 1 Line 21)
- The segmentation of the paper is arbitrary, such as Page 1 Line 18 to Line 27.And some of the sentences are not well expressed which makes me confused(like line 131 to 132).
- Question4 and 5 are not properly addressed, especially in conclusions content.

Author Response
The answer to revisor.
- Question 2 is not properly addressed. The specific setups of the experience need to be clearly specified.
To specify the setup were redrawn figure 4 added description (lines 131-145)
- When this passage studies the fatigue performance of the workpiece, does the influence of the material and defect of the workpiece been taken in consideration?
In this stage of research, the effect of the structure on the fatigue strength was not considered – only preliminary deformation.
- References citation 1 to 9 are still exist in your paper. (Page 1 Line 21)
It was fixed.
- The segmentation of the paper is arbitrary, such as Page 1 Line 18 to Line 27.And some of the sentences are not well expressed which makes me confused(like line 131 to 132).
These sentences were rephrased.
- Question 4 and 5 are not properly addressed, especially in conclusions content.
The discussion part and conclusion part were supplement and restructured.

Reviewer 5 Report
My comments in the manuscript have been taken into account and the manuscript has been enriched with additional necessary information.
Author Response
The list of editions.
- Lines 21-25 changes in reference’s citation.
- Line 42 changes in reference’s citation.
- Line 96 we have made changes to the table. The primary data were with some scatter in the chemical composition, current data have been updated in accordance with the certificate.
- Line 112 added information about the number of specimens.
- Redrawn figure 2 to eliminate the coincidence with our earlier article and to better understand it.
- Redrawn figure 3 for better perception.
- Lines 131-145 we have rewritten the description of the experiment as requested by the reviewer.
- To specify the setup, we were redrawn figure 4.
- Lines 161-163 stylistic edits.
- Line 213: part of the text has been moved to the methodology section.
- Lines 228-235 stylistic edits.
- Lines 245-248 stylistic edits.
- Lines 288-305, 323-329, 330-348: the discussion part and conclusion part were supplement and restructured.